# A Generic Model for Quantum Measurements

**DOI:** 10.3390/e21090904

**Published:** 2019-09-17

**Authors:** Alexia Auffèves, Philippe Grangier

**Affiliations:** 1Institut Néel, BP 166, 25 rue des Martyrs, CEDEX 9, F38042 Grenoble, France; 2Laboratoire Charles Fabry, IOGS, CNRS, Université Paris Saclay, F91127 Palaiseau, France

**Keywords:** quantum measurement, quantum foundations, algebraic quantum theory

## Abstract

In previous articles, we presented a derivation of Born’s rule and unitary transforms in Quantum Mechanics (QM), from a simple set of axioms built upon a physical phenomenology of quantization—physically, the structure of QM results of an interplay between the quantized number of “modalities” accessible to a quantum system, and the continuum of “contexts” required to define these modalities. In the present article, we provide a unified picture of quantum measurements within our approach, and justify further the role of the system–context dichotomy, and of quantum interferences. We also discuss links with stochastic quantum thermodynamics, and with algebraic quantum theory.

## 1. Introduction

Quantum measurements are present everywhere in current experimental physics, for instance when using individual qubits in quantum information research [1]. Such measurements may be strong or weak, and they may be implemented in a great variety of ways. To describe them theoretically, the main generic description remains the von Neumann model [2], where the system is first entangled with some ancilla, then a projective measurement is performed on the ancilla, and finally conclusions are drawn on the system’s state. Obvious examples are the Stern–Gerlach measurement for the spin (entanglement between the spin and the momentum, then between the spin and the position, and detection of the position), or Quantum Non Demolition (QND) measurement of a qubit state (entanglement with another “ancilla” qubit, and then direct detection of the ancilla).

However this “two-step” quantum measurement process has been controversial since the beginning of Quantum Mechanics (QM) because its second part (the projection) breaks the reversible unitarity of the evolution that would be expected from Schrödinger’s equation: This is the famous “collapse of the wave-function” that intrinsically binds together quantum measurement and irreversibility. To avoid breaking this reversible unitarity of quantum evolution, De Witt [3] in 1970 proposed to eliminate the second step altogether, and to consider a “universal wave function”, where a quantum measurement is not associated with a projection and a single result, but rather with a “branching of universes” inspired by Everett’s approach [4]. Before or after DeWitt’s proposal, there were many other attempts [5] to solve this measurement problem, and to deal with the so-called “quantum randomness” that comes with it [6,7].

Along this way, we introduced recently another approach called CSM like Contexts–Systems–Modalities [8]. The CSM approach takes just the opposite direction: rather than desperately seeking to explain why a quantum measurement provides a single observed result, we start from the obvious empirical evidence that it does. Our basic postulate is that a quantum measurement, performed on a given (quantum) system by a given (classical) apparatus, provides a unique, actual and objective outcome [9]. Obtaining this outcome is a physical phenomenon, with two essential features: (1) it will occur again and again with certainty if the measurement is repeated on the same system with the same setup, called a context, and (2) it may stochastically change when the context is changed. Our point of view differs significantly from the standard (von Neumann’s) approach, especially by the following aspects, first physical (i) and (ii) and then more formal (iii):(i)the certain and repeatable outcome obtained after a measurement manifests the existence of a physical “state”. For CSM, this state does not characterize the sole system as usually considered, but it is attributed jointly to the system and to the specified context, including e.g., the orientation of polarizers or magnets. To underline this difference, we call this “state” a modality.(ii)our postulates (see Section 2) imply a “contextual quantization”: a given measurement provides a modality (for a given system and context) among a quantized set of mutually exclusive [10] modalities. In another context, there will be another set of mutually exclusive modalities, but in general these new modalities will **not** be mutually exclusive with the previous ones, but incompatible [11]. The existence of non-exclusive (or incompatible) modalities is a specific non-classical feature resulting from contextual quantization, and it appears here before the introduction of any mathematical formalism.(iii)in this framework, we can demonstrate that the relation between modalities in different contexts can only be probabilistic [7], i.e., that quantum randomness is a necessary consequence of quantization and contextuality [8]. We also obtain Born’s rule, by a reasoning connected to Gleason’s theorem [12,13]. Unitary transformations do appear, but only to connect modalities, and they never “develop” to include the context.
Thus, the CSM approach is built upon the certainty and repeatability of quantum measurements in a given context. One may still ask for an explicit interpretation of quantum interferences, and, for the real meaning of the separation between system and context, despite the fact that they are both parts of the same physical world. In order to answer these questions, the purpose and the outline of the present article are the following:-In Section 2, we interpret quantum interferences within the CSM approach. Instead of being seen in the usual way as the superposition of different quantum states of the system, quantum interferences are shown to occur in reversible changes of contexts preserving the initial modalities, as opposed to irreversible changes of contexts realizing new modalities.-In Section 3, we provide a unified picture of quantum measurements within the CSM approach. We shall focus on QND (or ancilla-assisted) measurements, and show that CSM can describe strong or weak measurements, provided that any (small or large) quantum system is surrounded by an appropriate context. Measurement-induced irreversibility is then associated with intrinsically random jumps between modalities, in a picture very reminiscent of stochastic thermodynamics and quantum trajectories; this may have tangible consequences in the basic concepts in quantum thermodynamics [12,13,14].-Finally, in Section 4, we justify further the need of the system–context dichotomy. As explained in the articles quoted above, this dichotomy has the great virtue to eliminate the “wave-packet collapse” as it appears in the usual approach (because there is no wave-packet any more!), as well as any “instantaneous influence at a distance” in Bell test experiments [15,16,17,18]. Nevertheless, a remaining open issue is the physical “cut”, i.e., the physical separation between the system and the context. But something like a cut, usually called “sectorization”, does show up in more elaborate quantum algebraic formalisms, like C* algebras, which are used in quantum field theory to deal with infinite numbers of degrees of freedom, or in statistical physics to deal with the thermodynamic limit, i.e., with the limit where the number of particles becomes (mathematically) infinite [19]. We will show below that the algebraic formalism is suitable for our purpose, and can indeed manage quantum measurements, in a way compatible with our approach.

## 2. Interferences in the CSM Approach

### 2.1. Reminders

In previous works [12,13], we have shown that the quantum formalism (more precisely, the quantum way to calculate probabilities) can be deduced using a simple set of axioms built upon a physical phenomenology of quantization. Here, we just recall the basic features of our quantum realistic approach, called “CSM” for Contexts, Systems, Modalities. More detailed presentations are given in the references quoted above.

A context can be seen as a measuring device with given settings, which can be copied and broadcasted. It may be for instance an optical polarizer or a Stern–Gerlach magnet, in a given orientation defined classically. When coupled to a quantum system, the context gives rise to measurement outcomes, which are real, objective and actual phenomena. Our axioms are then the following:Axiom 1: Once measured on a given system in a given context, measurement outcomes are fully predictable and reproducible [20]. They are called modalities, and are attributed jointly to the (quantum) system and to the (classically defined) context.Axiom 2: For a given quantum system and classical context, there are *N* mutually exclusive modalities. The value of *N* is a property of the system, and is the same in all relevant contexts.Axiom 3: The changes between contexts have the structure of a continuous group (there is an identity corresponding to “no change”, the changes can be combined, and they have an inverse).

By construction, contexts, systems, and modalities are objective and real. The corresponding objectivity and realism are, however, quite different from their classical counterparts, and we call them “contextual objectivity” [21] and “quantum realism” [12]. Based on these axioms and on additional requirements for probabilities, we can show that [12,13]:The link between the *N* modalities in one context and the *N* (generally incompatible) modalities in another context must be probabilistic; this is a direct consequence of the quantization axiom 2.By associating a Hermitian rank-one N×N projector to each modality [22], the probability to get modality $v_{j}$ (associated with projector $P_{v_{j}}$) if one starts in modality $u_{i}$ (associated with projector $P_{u_{i}}$) is given by Born’s rule $p_{v_{j}|u_{i}}=\mathrm{Trace}\left(P_{v_{j}}P_{u_{i}}\right)$.A change of context is associated with a unitary matrix transforming the set of *N* mutually orthogonal projectors associated with the old context into a similar set associated with the new context.

To conclude this section, we point out that in CSM there is always one actual (or realized) modality, corresponding to a fully specified context. It is nevertheless useful to speak about other (non realized) modalities in other contexts, as possibilities that may occur with some probability. This makes the wording simpler, but within CSM there is no use of any counter-factual reasoning (see also discussion about Bell tests in Section 4.5).

### 2.2. Quantum Interferences in a Contextual World

A quantum interference is usually seen as the result of the coherent superposition of quantum states pertaining to a quantum system; the standard interpretation is thus essentially ignoring the context. However, quantum interferences can also be interpreted in the CSM perspective, building on the following observation: An “interference” is nothing but a certain and reproducible measurement outcome, in a properly defined context. In other words, an interference manifests the existence of a modality, i.e., a certain phenomenon in a given context that is probabilistically connected to some set of incompatible modalities in a different context. Therefore, an interference involves at least two contexts: The context that supports a modality, and any other different context, where the “quantum superposition” takes place.

These considerations shed new light on the transformation described above as a “change of context” that can be reversible (Axiom 3). To make the arguments explicit, let us consider a set $\left\{u_{i}\right\}$ of *N* mutually exclusive modalities in a given context, and another such set $\left\{v_{j}\right\}$ in another context. Since the $\left\{u_{i}\right\}$ and $\left\{v_{j}\right\}$ are non-exclusive (incompatible) modalities, there are $N^{2}$ probabilities $p_{v_{j}|u_{i}}$ for finding the particular modality $v_{j}$ (in the new context), when one starts with modality $u_{i}$ (in the initial context). Then, if one defines an a priori probability distribution (classical statistical mixture) $\left\{p\left(u_{i}\right)\right\}$ in the initial context, with $\sum_{i}p\left(u_{i}\right)=1$, the probability distribution in the new context is clearly
(1)$$p\left(v_{j}\right)=\sum_{i}\;p_{v_{j}|u_{i}}\;p\left(u_{i}\right),\;$$
where the distribution $\left\{p\left(u_{i}\right)\right\}$ may be itself the result of a change of context. For instance, if one goes from the context $\left\{u_{i}\right\}$ to $\left\{v_{j}\right\}$, and then back to the initial context, now denoted as $\left\{u_{k}\right\}$, the reverted form of Equation (Equation 1) gives
$$p\left(u_{k}\right)=\sum_{j}\;p_{u_{k}|v_{j}}\;p\left(v_{j}\right)$$
and, if the initial modality is known to be $u_{i}$, one has $p\left(v_{j}\right)=p_{v_{j}|u_{i}}$ so that the “return” probability is
(2)$$p_{\left\{v\right\}}\left(u_{k}|u_{i}\right)=\sum_{j}\;p_{u_{k}|v_{j}}\;p_{v_{j}|u_{i}}.$$

This relation tells that, after realizing a new modality in a new context $\left\{v_{j}\right\}$, returning back in the initial context will not deterministically bring back the initial modality: this is a consequence of the incompatibility of modalities between the two contexts. We will show below that this quantum irreversibility can be quantified by some genuinely quantum entropy production.

On the opposite, a reversible change of context should lead to record $p_{u_{k}|u_{i}}=\delta_{ki}$, as expected for mutually exclusive modalities if no context change happened. This is in agreement with quantum experiments which tell us that, as long as the modalities in the new context are not properly recorded, a context change is reversible, as if it did not happen at all. Such a change of context is simply described by reversible unitary transformations, leading to no entropy production. This is typically the case in an interferometer, as illustrated schematically in Figure 1.

Summarizing, going from context $\left\{u_{i}\right\}$ to context $\left\{v_{j}\right\}$, and then coming back to the initial context $\left\{u_{i}\right\}$, can be done in two different ways:an irreversible way, with a strictly positive entropy production, where the conditional probability $p_{\left\{v\right\}}\left(u_{k}|u_{i}\right)$ is given by Equation (Equation 2): this corresponds to realizing one modality in the context $\left\{v_{j}\right\}$, and we say that the measurement {v} has been performed.a reversible way, with no entropy production, where the conditional probability is given by $p_{u_{k}|u_{i}}=\delta_{ki}$: in this case, none of the modalities in the context $\left\{v_{j}\right\}$ was realized; in other words, the measurement has not been performed [23].

As said above, the equation for picking up a particular probability is given by Born’s formula:(3)$$p_{v_{j}|u_{i}}=p_{u_{i}|v_{j}}=\mathrm{Tr}\left(P_{v_{j}}P_{u_{i}}\right)={\left|\langle v_{j}|u_{i}\rangle \right|}^{2},$$ where we used the standard Dirac bra-ket notations. The conditions about these reversible and irreversible ways can then be rewritten more explicitly. In the reversible case, one has $p_{u_{k}|u_{i}}=\mathrm{Tr}\left(P_{u_{k}}P_{u_{i}}\right)=\delta_{ki}$ and thus
(4)$$\begin{array}{lll} p_{u_{k}|u_{i}} & = & |\langle u_{k}|u_{i}\rangle {|}^{2}\\ & = & |\sum_{j}\;\langle u_{k}|v_{j}\rangle \langle v_{j}|u_{i}\rangle {|}^{2}. \end{array}$$

In the irreversible case, one obtains from Equation (Equation 2)
(5)$$\begin{array}{lll} p_{\left\{v\right\}}\left(u_{k}|u_{i}\right) & = & \sum_{j}\;\mathrm{Tr}\left(P_{u_{k}}P_{v_{j}}\right)\;\mathrm{Tr}\left(P_{v_{j}}P_{u_{i}}\right)\\ & = & \sum_{j}\;{\left|\langle u_{k}|v_{j}\rangle \right|}^{2}{\left|\langle v_{j}|u_{i}\rangle \right|}^{2}. \end{array}$$

Equation (Equation 4) corresponds to adding probability amplitudes associated with “virtual paths” through the modalities $v_{j}$, whereas Equation (Equation 5) corresponds to adding probabilities associated with “real” modalities $v_{j}$. We have therefore the usual interference effects, not associated with a “wave function”, but only with the projective structure of the quantum probability law.

To conclude, going through an intermediate context can be done in two different ways: an irreversible way characterized by the realization of modalities of the intermediate context, and where the corresponding probabilities must be added, and a reversible way, where no intermediate modality is realized. In this latter case, the probability amplitudes corresponding to different possible intermediate modalities must be added, giving rise to quantum interferences. This is also reflected in the mathematical modeling of the transformation: The first case is characterized by a stochastic jump from one modality to another one, leading to a strictly positive entropy production that can be computed with the tools of stochastic thermodynamics (see below). In the second case, all context changes are described by unitary transforms that are reversible, evidencing that the CSM and standard pictures are operationally consistent.

## 3. Contextual Approach of QND Measurements and Irreversibility

### 3.1. A Simple Quantum Measurement Model

We shall now introduce a simple model in which the two cases presented above simply appear as extreme situations. Namely, we still assume that the system is initially prepared in a modality $|u_{i}\rangle$ of some initial context $C_{u}$, but that the measurement with respect to the intermediate basis $\left\{|v_{j}\rangle \right\}$ is performed through a second quantum system, called the “meter” or “ancilla”. Hence, the process we shall model corresponds to an ideal indirect measurement according to Von Neumann [2], or in a modern terminology of a Quantum Non Demolition (QND) measurement [24]: The system and the meter are first entangled, and then the meter is measured within a fixed context $C_{m}$ whose modalities are denoted $\left\{|x_{l}^{m}\rangle \right\}$.

In a QND measurement, the system–meter compound is initially prepared in the modality $|u_{i},x_{1}^{m}\rangle$, giving rise to a reproducible outcome if the system is measured in the $C_{u}$ context. Then, the system–meter coupling gives rise to a global change of context for the compound, described by a standard unitary transformation:$$\begin{array}{l} |u_{i},x_{1}^{m}\rangle =\sum_{j}\langle v_{j}|u_{i}\rangle |v_{j},x_{1}^{m}\rangle \\ \to |\xi_{i}\rangle =\sum_{j}\langle v_{j}|u_{i}\rangle |v_{j},w_{j}^{m}\rangle , \end{array}$$ where we have used the closure relation $\sum_{j}|v_{j}\rangle \langle v_{j}|=\hat{1}$. Depending on the strength of the system–meter coupling, the meter modalities $\left\{|w_{j}^{m}\rangle \right\}$ are not necessarily orthogonal, i.e., they do not necessarily represent a set of new modalities in a new context. In the spirit of the previous section, we can investigate the nature of the context change experienced by the system in this extended framework. Explicitly, getting the modality $|u_{k}\rangle$ when going back in the initial context $C_{u}$ after the system has interacted with the meter comes with the probability:(6)$$\begin{array}{lll} P(u_{k}) & = & \langle \xi_{i}|P_{u_{k}}⊗{\hat{1}}_{meter}|\xi_{i}\rangle \\ & = & \sum_{l}\langle \xi_{i}|u_{k},x_{l}^{m}\rangle \langle u_{k},x_{l}^{m}|\xi_{i}\rangle \\ & = & \sum_{j,j^{\prime }}\langle u_{i}|v_{j}\rangle \langle v_{j}|u_{k}\rangle \langle w_{j}^{m}|w_{j^{\prime }}^{m}\rangle \langle u_{k}|v_{j^{\prime }}\rangle \langle v_{j^{\prime }}|u_{i}\rangle . \end{array}$$

Two limiting cases thus emerge. If the $|w_{j}^{m}\rangle$ are orthogonal—namely, if they correspond to a set of exclusive modalities for the meter, then $\langle w_{j}^{m}|w_{j^{\prime }}^{m}\rangle =\delta_{jj^{\prime }}$ and thus Equation (Equation 6) reduces to Equation (Equation 5). Therefore, measuring one of the meter’s modalities in $C_{m}$ boils down to measuring one system’s modalities in the intermediate context $C_{v}$. Just like above, this gives rise to irreversibility of purely quantum nature (see below). On the opposite, if $\langle w_{j}^{m}|w_{j^{\prime }}^{m}\rangle =1$, i.e., if the $|w_{j}^{m}\rangle$ are indistinguishable, then the measurement on the meter has no effect on the system, and one gets Equation (Equation 4), which is equivalent to a reversible change of context where the initial modality is conserved. The intermediate situation is illustrated in Figure 2. Note that the results associated with the meter do not have to be read: provided that the interaction system–meter is such that the meter modalities are orthogonal, the system’s modalities will be defined in the context corresponding to the $\left\{v_{j}\right\}$ modalities. Therefore, this context is a “pointer basis” according to the usual theory of decoherence [25,26,27].

Summarizing, we have recovered in the CSM framework that the result of a measurement performed through the meter is related to the quantum distinguishability of the meter modalities, and continuously goes from null to projective. Interestingly, the same continuity appears as far as the reversible/irreversible nature of the measurement is concerned: As we will show now, the entropy produced by the measurement is null if the measurement is reversible and equals the (post-measurement) system’s Shannon entropy if the measurement is projective.

### 3.2. Quantitative Assessment of Irreversibility

Quantum stochastic thermodynamics [14] provides a convenient framework to define the so-called entropy production that relies on the notion of stochastic quantum trajectory. In the CSM language, a quantum trajectory is nothing but a time-ordered sequence of modalities obtained by performing measurements within a sequence of contexts $C(t_{k})$. In the present situation, stochasticity stems from projective measurement. We show in Appendix A that the mean entropy production, averaged over the trajectories, is simply the Shannon’s entropy associated with the distribution of measurement outcomes of the final measurement.

This allows us to specify the conditions to obtain a reversible change of context. Following the prescription of stochastic thermodynamics, a transformation is reversible iff entropy production is null on average, which can only be reached if it is null for each trajectory. In the case of QND measurements performed through a meter, reversibility can be reached in the case of weak measurements where the quantum states of the meter are asymptotically not distinguishable. This shows that quantum interferences associated with reversibility, as well as the realization of new modalities associated with irreversibility, do fit in our generic picture for quantum measurements; actually, a full continuum of situations is possible between the two extreme cases presented in Figure 1.

## 4. An Algebraic Formalization of Contexts within CSM

### 4.1. The Measurement Problem in QM

In the CSM framework, the structure of quantum measurements is directly defined by the basic axioms, so, if these axioms are accepted, there is no “measurement problem” at all. Nevertheless, in the previous section, we have been using standard Dirac notations for (pure) quantum states, which are re-interpreted as modalities by CSM, and thus require a context for being defined. Then, a natural question shows up: where is the context in the quantum formalism we have been using?

A first answer is that the values of the relevant context parameters (e.g., the polarizer’s angle) are included for the definition of the observables (Hermitian operators), whereas a modality corresponds to eigenvalues (e.g., transmitted or not) associated with a given eigenstate. In order to define the modality, the eigenvalues are not enough: one also has to specify the relevant observable (polarizer’s angle), or, more generally, the relevant complete set of commuting observables (CSCO), i.e., the context.

Thus, a modality is rightfully attributed to a system (carrying measured eigenvalues), within a context (carrying the parameters which define the relevant CSCO). The eigenstate itself appears only as a second step, as a projector associated with an equivalence class of modalities: this is how Born’s rule comes out in CSM [13]. However, a frequently asked question is: the context is made also with atoms, like the system, so why don’t these atoms show up somewhere? In addition, what are the “relevant context parameters” with respect to the atoms of the context? In order to answer these questions, we make a step backward, and come back to standard QM.

### 4.2. From the Measurement Problem to Infinities in QM

The usual textbook presentation of QM is largely based on John von Neumann’s famous opus [2], using the theoretical description already quoted: the system is first entangled with some ancilla, then a projective measurement is performed on the ancilla, and finally conclusions are drawn on the system’s state. The first step (entanglement with some ancilla) has been spelled out in detail in the previous section. The second step (projection postulate) is required to get a consistent account of a quantum measurement, so it has to be an intrinsic part of the theoretical construction. However, it is often said that this second step breaks the unitary evolution expected from Schrödinger’s equation, from which the entangled state $|\xi_{i}\rangle =\sum_{j}c_{ji}\;|v_{j},w_{j}^{m}\rangle$ of Section 2.3 should become
(7)$$|\xi_{i}\rangle =\sum_{j}c_{ji}\;|v_{j},w_{j}^{m_{1}},w_{j}^{m_{2}},...,w_{j}^{m_{M}}\rangle ,$$
with *M* going to infinity as more and more atoms (and photons) of the context are involved. This is obviously different from the post-measurement density matrix
(8)$$\rho_{i}=\sum_{j}|c_{ji}{|}^{2}\;|v_{j}\rangle \langle v_{j}|⊗\rho_{j}^{m},$$
where $\rho_{j}^{m}$ is a large density matrix involving the whole context, organized in such a way to measure and register the result *j* of the measurement.

All physicists will agree that (Equation 8) should be used for all practical purposes after a quantum measurement, but deciding between (Equation 7) and (Equation 8) “at the fundamental level” has been a long and harsh debate in physics. Two extreme positions are the one defended by DeWitt in [3], where only (Equation 7) is valid, vs. the one resulting from von Neumann’s projection postulate, where only (Equation 8) is valid. In the case where *M* is very large but finite, an explicit calculation of the system–apparatus interaction tells that, after a very short relaxation time, (Equation 8) is valid within an excellent approximation [6]. The same conclusion is obtained if one considers that some particle in (Equation 7) flies away from the observation region; then, (Equation 8) is obtained from a partial trace [26,28]. However, the question might be ultimately a mathematical one: since (Equation 7) is clearly valid during the first steps of the measurement, when only a few ancillas are entangled with the system as described in Section 2, whereas (Equation 8) appears later on when *M* becomes very large, what is the correct way to take the limit M→∞?

### 4.3. The Algebraic Approach

To answer this question, it should be noted that von Neumann himself did not like the projection postulate, and he looked for another approach. A trend of his efforts is given in an article published in 1939, “On infinite direct products” [29]. This research was apparently slowed down by World War 2 [30], but the work by von Neumann and others finally lead to what is known as the algebraic approach to QM, abbreviated here by AQM. Popular accounts of AQM tell that its goal is to describe infinite dimensional systems, as they occur for instance in quantum field theory, or at the “thermodynamic limit” of quantum statistical mechanics. It is therefore relevant for our question about M→∞ [19].

In his 1970 article [3], DeWitt knew quite well about AQM, but he dismissed it in two sentences: *“Despite the undeniable elegance and importance of the C*-algebra approach to statistical mechanics, none of us has even seen an infinite gas. In addition, the real universe may, in fact, be finite.”* In our opinion, this is a serious misunderstanding of the whole issue, with many undesirable consequences. The formalism of AQM is very mathematical, and, when doing mathematics, there is no problem in manipulating infinite quantities, countable or not—no need to see an infinite gas for doing that. However, now quoting Hilbert [31], *“Our principal result is that the infinite is nowhere to be found in reality. It neither exists in nature nor provides a legitimate basis for rational thought—a remarkable harmony between being and thought. (...) Operating with the infinite can be made certain only by the finitary.”* Thus, when applying AQM to physics, the purpose of the theory is **not** to consider “really infinite” quantities, but to deal with arbitrarily large quantities, and to warrant consistency of the calculations, from both a physical and a mathematical point of view. Considering that the number of degrees of freedom of any actual measurement apparatus is extremely large, it is unwarranted that the elementary QM approach used by DeWitt is suited to what he purported to describe.

In practice, AQM is able to provide a consistent mathematical description of both finite and infinite systems, by looking at algebras of observables, and at associated Hilbert spaces. More precisely, it is known [29] that an infinite tensor product is problematic as an Hilbert space because its dimension is continuously infinite (like real numbers), and it is not separable any more. Therefore, rather than considering a non-manageable infinite tensor product, the overall algebra and associated states may be reorganized as a direct sum of unconnected super-selection sectors. Such a representation of an operator algebra is said to be reducible, or, equivalently, it is not a pure state [32]. On the other hand, each sector corresponds to an irreducible representation, i.e., to a pure state for the system, and to a unique measured value (or set of values) associated with this pure state [28]. This clearly gives (Equation 8) and not (Equation 7) as the proper M→∞ limit.

At the global level, this is in perfect agreement with the usual projection postulate, but now it is no more postulated “by fiat”: it appears as the mathematical consequence of extending the number of degrees of freedom to infinity during the measurement, reorganizing it as a direct sum of sectors, and looking at the result in one of these sectors. It must be emphasized that this is a **mathematical** description of the phenomenon under study, which does not give any ontological “branching of universes”, but simply a way to calculate the probability of the various measurements results, and to define the resulting pure system state, which is the irreducible representation post-selected by the observed result.

In another view [28], considering that the apparatus is not actually infinite, it is required to invoke particles escaping to infinity, in order to justify the super-selection sectors: they appear because remote parts of the environment are ignored. Such a point of view is close to environment-induced decoherence [26,27], but, in our opinion, it does not fully exploit the algebraic framework as a mathematical limiting case. The point is not to have a *physically infinite* number of particles, which would be meaningless, but, in order to assert that to describe the very large number of particles in a measurement context appropriately, it is required to be consistent with the limit of a *mathematically infinite* number of particles [33].

Anyway, given that many different approaches conclude like CSM that (Equation 8) rather than (Equation 7) is the correct post-measurement state, it seems safe to give an algebraic content to the CSM axioms under the form: *A modality is obtained in a given context when a superselection rule appears, splitting the different measurement results that are possible in this context*. This is obviously equivalent to the real change of context discussed in Section 2, consistent with CSM, and in agreement with other approaches, even if they support different ontologies [19].

### 4.4. Unscrambling Physics from Mathematics

Given that the algebraic formalism is a tool to calculate probabilities, what are the physical objects, properties and events that are described by these probabilities? A simple and consistent definition of these physical objects and events is that they are nothing but the quantum systems, measurement devices, and measurement results, in agreement with the CSM ontology [8]. Here, a modality is a fully predictable and reproducible measurement result, and it is associated with a pure state, or with an irreducible representation. Given an initial modality, a new one is obtained when changing the context, as a result of the actualization or “reading out” of the result (see Section 2). This is in agreement both with AQM and with the projection postulate, without any “reduction of the wave packet”, since there is no such object, neither in the ontology nor in the formalism. Thus, the non-unitarity of the evolution during a measurement basically comes from re-organizing the algebra of observables in disconnected (or superselected) sectors, each sector being ear-marked by the result observed in a well-defined new context, among the *N* possible mutually exclusive results.

Again, a crucial point in the CSM approach is that the modality/pure state/irreducible representation is attributed to both the quantum system and the classical context that is a generic name for the measurement apparatus. The actual context must be specified; otherwise, there is no way to decide how the re-arrangement will proceed: giving a density matrix is not enough to determine a context, but giving a context allows one to determine the density matrix (Equation 8). This is consistent with the CSM view that the purpose of the quantum formalism is to describe the behavior of physical objects, and not to replace them by mathematical objects.

### 4.5. Some Illustrations

As an illustration, let us quote here some often-heard but nevertheless erroneous statements.

*it has been proven that one can make quantum superpositions of large objects, like in interference experiments with big molecules, and this shows by extrapolation that everything can be put in a quantum superposition.* This claim is obviously wrong because an external context is always needed to observe the interference effect, as a modality jointly attributed to the system and the measurement context. In CSM, there is no intrinsic restriction in the physical size of the system, as long as there is a context around it, in order to define and observe the relevant modalities. For instance, there is no physical context able to distinguish the orthogonal “states” $(|living\rangle +\left|dead\rangle \right)/\sqrt{2}$ and $(|living\rangle -\left|dead\rangle \right)/\sqrt{2}$ for a Schrödinger cat (the animal), so such “states” are meaningless extrapolations. On the other hand, such quantum states may be meaningful for a metaphoric cat, e.g., a large quantum computer, provided that it is surrounded by the appropriate measurement context. Actually, small metaphoric cats made with a few photons have already been observed in many experiments—see, e.g., [34].*if one cannot make arbitrarily large quantum superposition, then quantum computing will have problems.* This claim is also incorrect, and it is forbidden is to make “infinite regression”, i.e., superpositions that would “swallow” the context. However, since the number *M* of qubits is not mathematically infinite, the exponential advantage provided by the $2^{M}$ dimensions of the Hilbert space is potentially useful. On the other hand, whichever *M* is in practice, an outer classical context (with $M_{context}\gg M_{system}$) is required to make sense of the results, within the usual macroscopic world—as it is in some sense obvious.*in CSM, there is no reference to any observer or agent, but the parameters of the context (e.g., the orientation of the Stern and Gerlach magnet) must be arbitrarily chosen by an agent; they have therefore a subjective character, so the observer is still there.* This is a counter-factual reasoning, which gives the same status to “what might have been made” and to “what has actually been made”. It is well known from loophole-free Bell tests [15] that counter-factual reasoning contradicts quantum empirical evidence: what matters in a Bell experiment are the parameters of the context that have actually been used, and, in a Bell test, no observer decides them in advance, since they are randomly chosen at the last instant [18]. What matters is the actual status of the context, determined by a physical independent choice, unrelated with the subjectivity of an observer [17]. Looking at the same issue on the mathematical side, we come to the same conclusion: the re-arrangement or re-organization of the algebra as a direct sum of irreducible representations requires that the actual parameters of the context are well defined—as they are at the macroscopic level.*if a split is introduced between systems and contexts, then QM is not universally valid.* It should be clear now why this statement is wrong: for being correct, it should write “then, type I QM is not universally valid”, according to the usual AQM type classification [19,28,29,30]. It is clear indeed that the usual type I QM formalism cannot properly manage the mathematical limit of an infinite number of particles, and is thereby doomed to fail.

## 5. Conclusions

As a conclusion, revisiting the measurement problem in the CSM framework leads us to a conclusion opposite to DeWitt’s [3]. By carefully avoiding extrapolations of the formalism of elementary QM, we avoid mixing up what belongs to physics and what belongs to mathematics. For our purpose, it is crucial to separate

-the mathematical formalism, which must be the algebraic one in order to manage properly the infinities, even if “the infinite is nowhere to be found in reality” [31];-the physical ontology, based on quantum systems, measurement devices (contexts), and reproducible measurement results (modalities), which turns out to be in excellent correspondence both with the empirical evidence and with the algebraic mathematical framework.

We emphasize that we don’t expect the formalism to provide any “emergence” of classical reality. For CSM, the classical reality is already there, embedded in the context which is part of the initial axioms. Then, the problem is to look for a formalism able to describe both the micro and macro levels (separated by a “cut”) in a mathematically and physically consistent way.

Thus, CSM has a minimalistic ontological content, based on empirical evidence at the macroscopic level, and including all quantum objects that have been discovered by physics. It makes sense within the idea that a real world exists independently of the observer, and that one can make contextually objective statements about it [21]. Our purpose is not to explain “why” we see things as we do, from a theory of emergence; it is rather to consistently explain “how” the world of objects and events behaves, based on its underlying quantum structure. This fits also with our preferred slogan, “quantum mechanics can explain anything, but not everything” [35].

This work may be pursued in various directions, e.g., for reconsidering quantum non-locality in Bell tests [16,17], quantum randomness [7], and Born’s rule [12,13]. It also has a pedagogical value, to unlock some barriers between mathematical and physical issues in quantum mechanics, and to clarify what belongs to each realm; this has not always been very clear throughout the history of QM.

## Figures and Tables

**Figure 1 entropy-21-00904-f001:**
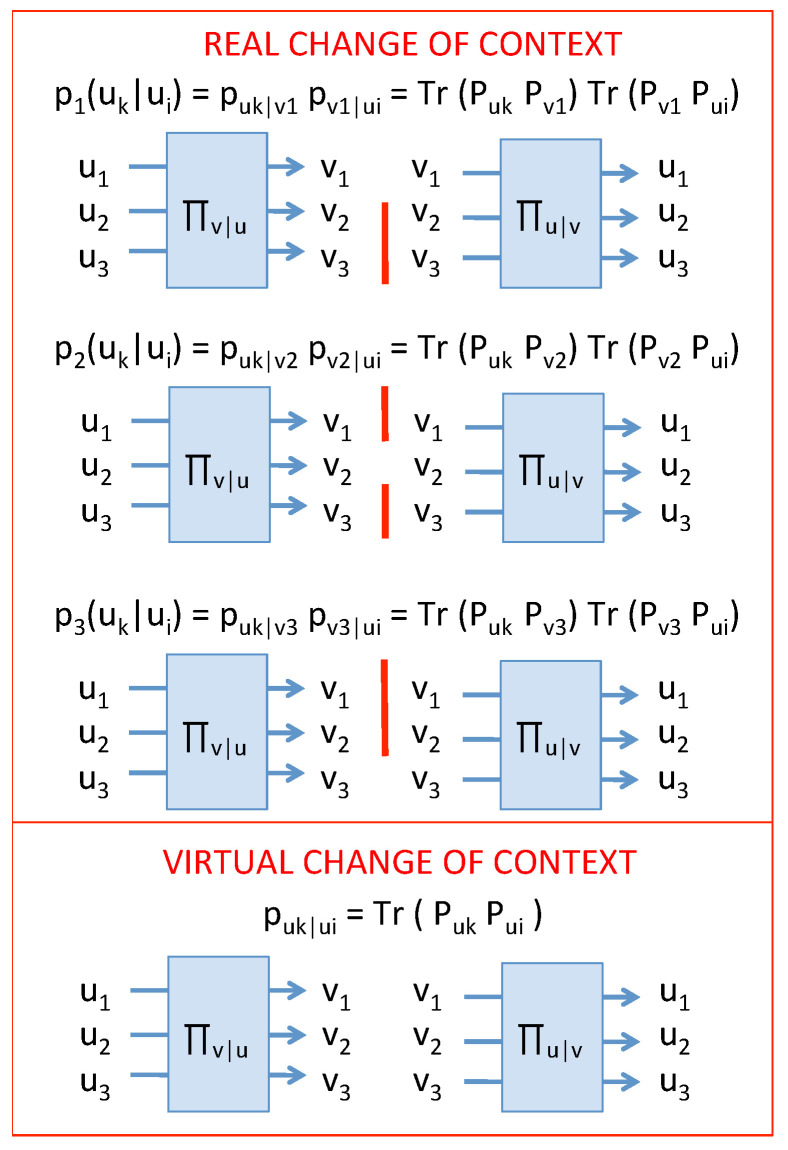
Two possible ways to change the context and come back. An irreversible change of context is signaled by the realization of one modality $v_{j}$ in the new context. This phenomenon eliminates all other possible new modalities $v_{i\neq j}$ because modalities are mutually exclusive in a given context. If the outcomes are not read, one has to sum probabilities over the N possibilities, and one gets $p(u_{k}|u_{i})$ given by Equation (Equation 2). As the opposite extreme case, a reversible change of context does not give rise to the realization of any new modality in the context $\left\{v_{j}\right\}$. In that case, one has $p_{u_{k}|u_{i}}=\delta_{ki}$, corresponding to a sum of probability amplitudes (see text). The above figure in the reversible case can be seen as an interferometer, where the $\left\{v_{j}\right\}$ modalities correspond to “which-path” results.

**Figure 2 entropy-21-00904-f002:**
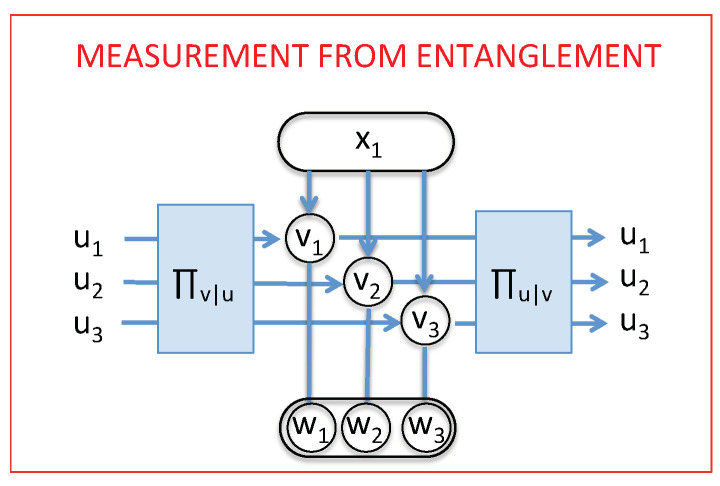
General way to change the context and come back, by entangling the initial system with another system, considered as the meter. To recover the previous cases, the system–meter interaction must be a QND interaction in the context $\left\{v_{j}\right\}$ (see text). Depending on the strength of this interaction, one can recover real, virtual, and “weak” measurements.

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
