# Peer review of "A Generic Model for Quantum Measurements"

_entropy, 2019, doi:10.3390/e21090904_

Round 1

Reviewer 1 Report

In their manuscript, the authors provide a new axiomatic approach to quantum mechanics called CSM (contexts-systems-modalities), in an attempt to get rid of some issues related to the measurement problem in quantum mechanics.

The paper is fluently and nicely written, the topic definitely interesting and the type of approach quite appealing, considering the outcome of a measurement as a foundational element for the formulation of the theory and not viceversa.

In my opinion, the paper is worth publishing, given that the authors fix some points.

For example, in the introduction the authors write: "Our basic postulate is that a quantum measurement, performed on a given (quantum) system by a given (classical) apparatus, provides a unique, actual and objective outcome.".

This is not only in contrast with many empirical evidences (unless we are talking about eigenstates of the measured observable), but also with the first axiom of their derivation, stating "Once measured on a given system in a given context, measurement outcomes are fully predictable and reproducible. They are called modalities, and are attributed jointly to the (quantum) system and to the (classically defined) context." (and thus referring to a post-wavefunction-collapse eigenstate-like situation). The authors should clarify this crucial concept, here as well as in other parts of the manuscript.

Furthermore, the bibliography should be extended a bit, specially considering that now about 30% of the references are self-citations.

Author Response

Answer to Referee 1

In their manuscript, the authors provide a new axiomatic approach to quantum mechanics called CSM (contexts-systems-modalities), in an attempt to get rid of some issues related to the measurement problem in quantum mechanics.

The paper is fluently and nicely written, the topic definitely interesting and the type of approach quite appealing, considering the outcome of a measurement as a foundational element for the formulation of the theory and not viceversa.

In my opinion, the paper is worth publishing, given that the authors fix some points.

We thank the referee for his positive assessment of our ms. 

For example, in the introduction the authors write: "Our basic postulate is that a quantum measurement, performed on a given (quantum) system by a given (classical) apparatus, provides a unique, actual and objective outcome.". This is not only in contrast with many empirical evidences (unless we are talking about eigenstates of the measured observable), but also with the first axiom of their derivation, stating "Once measured on a given system in a given context, measurement outcomes are fully predictable and reproducible. They are called modalities, and are attributed jointly to the (quantum) system and to the (classically defined) context." (and thus referring to a post-wavefunction-collapse eigenstate-like situation). The authors should clarify this crucial concept, here as well as in other parts of the manuscript.

We thank the reviewer for asking this clarification. Actually these two statements are not at all in conflict, but they tell different things :

- the «  basic postulate » tells that very generally, there is only one (possibly random) outcome to any  given quantum measurement, performed on a given (quantum) system by a given (classical) apparatus.  This statement is equivalent to postulating that  there is a single macroscopic world, or also to excluding counterfactual considerations about a measurement. This statement is obviously in agreement with empirical evidence, but its role is nevertheless useful, in order to distinguish our approach from other ontological views, such as many-worlds. An immediate consequence is that  we don’t have to « demonstrate the emergence of a single result », since it is our starting point. 

- the first axiom comes *in addition* to the basic postulate, and it tells something essential about repeatability : « Once measured on a given system in a given context, measurement outcomes are fully predictable and reproducible »  (in usual formal terms, they are eigenstates of the measured observable). So not only there is only one outcome (basic postulate), but also this outcome is (ideally) repeatable as long as the system and context are kept the same. 

We have modified a sentence (shown in red) and added a footnote [9] to clarify these points. 

Furthermore, the bibliography should be extended a bit, specially considering that now about 30% of the references are self-citations.

We have added references [20] and [29] (in red), and the ratio is now below 30%.  We do acknowledge that this article is not a review, and given the huge literature on quantum measurement, the present article is centered on presenting our approach. We also underline that since our interpretation is quite recent, it is of course not as known as the many-worlds interpretation can be. We thus have to include self citations in order not to reproduce all the material we have already published and that is necessary to the understanding of our new manuscripts.

Author Response

Answer to Referee 2

This article is part of a series on a theory or interpretation of quantum mechanics. As such it concerns foundations of quantum mechanics. This is a contentious and in my mind very much unresolved question I personally don't think we understand quantum mechanics, which is a very disturbing point of view to have! 

Response: We thank the referee for his comments. The feeling of a lack (or incompleteness) in the understanding QM is shared by many physicists, and it is certainly also a motivation for our work. 

I have read the present submission, and also the authors' paper on coming to terms with the Bell nonlocality analysis of EPR, Ref. 19. I emphasize the latter because I think it is the most difficult aspect of quantum mechanics to deal with.

The authors take an approach which they call CSM for "Contexts-Systems- Modalities." As best I can understand it, this approach is much like the "orthodox" Copenhagen view, in which there is a split between observer or system. However, the authors want to take a "realist" view, in distinct contrast to Copenhagen, which seemingly tries to be as anti-realist as possible.

Response: So far we agree with the general statements by the referee. 

I have never been comfortable with the Copenhagen view, with its split between (quantum) system and (classical) observation, which I find incomprehensible in view of the Schrodinger time evolution of the total system, which demands a seamless continuity of evolution of all elements of the total system. So I am sympathetic to attempts to find an alternative. Examples of such alternatives are many worlds, Bohmian, spontaneous collapse. The first two are based on the Schrodinger time evolution, with some added features in the Bohmian approach.

With respect to the present paper, I unfortunately do not see that the authors are really making headway at getting past the Copenhagen approach. They don't seem to take into account explicitly the Schrodinger time evolution. I don't see how their approach really is more "realist" than Copenhagen. They just seem to be sweeping the questions under the rug, so to speak, the rug being their formalism.

Response: The basic issue with our approach is not  so much the formalism, it is rather what we may call the ontology; sorry for this word which sounds weird to physicists, it simply means that we want to define first physical objects and physical properties, *before* introducing the formalism. Since we speak about QM, it is not a big surprise that these physical definitions imply quantization, and contextuality. Then we can give objective properties (modalities) to physical objects which associate a system and a context. In our opinion this is a (non-classical) realist approach, and it is not swept under the rug, but rather explicitly stated as the starting point. 

Furthermore, with respect to Bell-EPR, which again, seems to me like the biggest problem with understanding quantum mechanics I don't at all see that they have gotten rid of the "spooky action at a distance" that so famously bother Einstein. Again, it seems to me they are just throwing the real problem under the rug. The lesson or logic of EPR is still there.

Response: We perfectly agree that the lesson or logic of EPR is still there, and it is not our purpose to « get rid of it ». In our view, the ultimate lesson to be drawn from EPR is that classical physics and quantum physics are built up in a quite different ways. What is « specifically quantum » is contextual quantization, related  to contextual objectivity, ie to the idea that well defined physical properties don’t belong locally to the system, but belong to the system within a context. This embeds quantum non-locality and Bell’s inequality violation, without any spooky action at a distance, as explained in ref. 22 (previously 19).  In some sense, there is no free lunch : we speak about quantum realism, and this is certainly not classical realism, but still this is a meaningful realism. We have also added ref. 20 about another contextual view on this issue, and ref. 21 about a short article on the Einstein-Bohr debate within our approach. 

Unfortunately, therefore, I am not enthusiastic about this paper. That does not mean I am totally negative. As I say, the shaky foundations of quantum mechanics are there for all to see. They have been that way for something like a century now. Many people have worked on this; I don't see that anyone has totally succeeded. The authors have clearly thought about what they are doing, and have a track record in presenting their point of view. If there is enthusiasm for the work in the present paper from other reviewer(s) than myself, I would not want to prevent publication. Perhaps this approach will lead somewhere, though I am unlikely to become a follower.

Response: We are aware that this quantum puzzle has been discussed for almost one century, and it would be pretentious to declare solving it at once.  We just want  to throw a few white stones in a hopefully interesting direction, and will see what comes out of this.

Reviewer 3 Report

The paper describes a model for quantum measurements in the framework of the CSM (Context-Systems-Modalities) approach introduced recently by the same authors. I am an experimental physicist with very limited ability in mathematical subtleties but I find no faults in the authors approach. What I find particularly interesting is section 4 where the authors put forward a few examples to better illustrate their approach. This substantially helps a reader like myself more concerned with the physical meaning of the CSM approach.

Even if I may have doubts on the usefulness of a new foundational approach to QM I do not believe this is the question here. The paper describes the interpretation of measurements in quantum mechanics from a novel approach. It does it correctly, consistently and, more importantly in a way bound to interest the reader. For this reason I recommend publication. 

Author Response

Answer to Referee 3

The paper describes a model for quantum measurements in the framework of the CSM (Context-Systems-Modalities) approach introduced recently by the same authors. I am an experimental physicist with very limited ability in mathematical subtleties but I find no faults in the authors approach. What I find particularly interesting is section 4 where the authors put forward a few examples to better illustrate their approach. This substantially helps a reader like myself more concerned with the physical meaning of the CSM approach.

Even if I may have doubts on the usefulness of a new foundational approach to QM I do not believe this is the question here. The paper describes the interpretation of measurements in quantum mechanics from a novel approach. It does it correctly, consistently and, more importantly in a way bound to interest the reader. For this reason I recommend publication. 

We thank the referee for his positive assessment of our ms. Both authors have been doing experiments at some stage of their career, so we are quite attentive to feedback  from experimentalists; and actually, the CSM ideas are not so far from the « common sense » used in physics labs.